# CROSS-MODAL DOMAIN ADAPTATION FOR REINFORCEMENT LEARNING

## ABSTRACT

To overcome the unbearable reinforcement training of agents in the real-world, the sim-to-real approach, i.e., training in simulators and adapting to target environments, is a promising direction. However, crafting a delicately simulator can also be difficult and costly. For example, to simulate vision-based robotics, simulators have to render high-fidelity images, which can cost tremendous effort. This work aims at learning a cross-modal mapping between intrinsic states of the simulator and high-dimensional observations of the target environments. This cross-modal mapping allows agents trained on the source domain of state input to adapt well to the target domain of image input. However, learning the cross-modal mapping can be ill-posed for previous same-modal domain adaptation methods, since the structural constraints no longer exist. We propose to leveraging the sequential information in the trajectories and incorporating the policy to guide the training process. Experiments on MuJoCo environments show that the proposed cross-modal domain adaptation approach enables the agents to be deployed directly in the target domain with only a small performance gap, while previous methods designed for same-modal domain adaptation fail on this task.

## 1 INTRODUCTION

Deep Reinforcement Learning (DRL) for vision-based robotic-control tasks has achieved remarkable success in recent years (Francis et al., 2020; Zhang et al., 2019; Zeng et al., 2018; Riedmiller et al., 2018; Levine et al., 2018). However, current RL algorithms necessitate a substantial number of interactions with the environment, which is costly both in time and money on real robots. An appealing alternative is to train policies in simulators, then transfer these policies onto real-world systems (Rao et al., 2020; James et al., 2019; Yan et al., 2017).

Due to inevitable differences between simulators and the real world, which is also known as the "reality gap" (Jakobi et al., 1995), applying policies trained in one domain directly to another almost surely fail, especially in visual-input tasks, due to the poor generalization of RL polices (Cobbe et al., 2019). Domain adaptation is a common way to improve transferability by mapping inputs from two domains to an aligned distribution. Although distribution alignment is difficult with limited data, many recent works have adopted unsupervised visual domain adaptation (Hoffman et al., 2018; Ganin et al., 2017; Yi et al., 2017; Kim et al., 2017) to learn the mapping function without a ground-truth pairing. These adaptation methods exploit structural constraints (Fu et al., 2019) in two same-modal domains (i.e., learned on simulated **images** and deployed on real **images**) to overcome the intrinsic ill-posedness of distribution matching as shown in Fig. 1(a) — mapping an instance in the target domain to anything of a similar probability in the source domain is "reasonable" if we only consider distribution matching.

However, training on simulated images introduces unwanted costs and difficulties, which are ignored in current works. First, a rendering engine needs more human engineering and runs much slower (can be up to $20\times$ slower according to Xia et al. (2018)) than a pure rigid body simulator, which adds considerable cost to the overall process. Second, using RL methods to train a policy with image inputs is usually harder than training with state inputs (Kaiser et al., 2020; Tenenbaum, 2018), resulting in a sub-optimal simulation policy. An ideal solution to avoid such problems is to train policies with simulated states and adapt the learned polices to real-world images. However, all

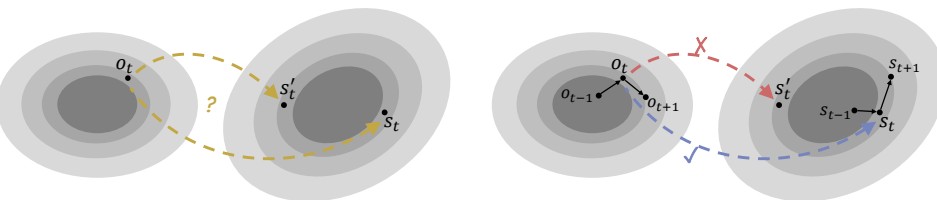

(a) Mapping without Sequential Structure      (b) Mapping with Sequential Structure

Figure 1: Illustration of mapping functions with and without sequential structure from the target domain (left) to the source domain (right). Shaded regions denote data distributions, where the darker the color, the higher the probability. In Fig. 1(a), both $s_t$ and $s_t'$ are "realisic" source domain instances, but only $s_t$ is correct. Since they are of similar probabilities, distribution matching may map $o_t$ to any of them. In RL, the policy may output unreliable actions when taking these incorrectly mapped states as inputs. In Fig. 1(b), a sequential structure can help rule out the wrong mapping based on trajectory contexts.

the structural constraints based on the modality consistency can not be used and the distribution alignment task by learning a mapping function becomes hard to solve.

In this paper, we propose Cross-mOdal Domain Adaptation with Sequential structure (CODAS) that learns a mapping function from images in the target domain to states in the source domain. With the help of the learned mapping function, policies trained on states in the source domain can be deployed in the target domain of images directly. Specifically, based on the sequential nature of reinforcement learning problems, we formulate the cross-domain adaptation problem as a sequential variational inference problem and derive a series of solvable optimization objectives in CODAS. It is worth noting that our work is different from recent works that learn state embeddings from image inputs, which map images to an arbitrary subspace in a low-dimensional vector space. In CODAS, we embed the image space into a vector space with clear meanings (defined in the state-based simulator), which improves the interpretability of the policy when deployed in the real world.

We evaluate our method on 6 MuJoCo (Todorov et al., 2012) environments provided in OpenAI Gym (Brockman et al., 2016), where we treat states as the source domain, and rendered images as the target domain, respectively. Experiments are conducted in the scenario where only offline real data are available. Experiment results show that the mapping function learned by our method can help transfer the policy to target domain images with a small performance degradation. Previous methods that use unaligned Generative Adversarial Networks (GANs) suffer from a severe performance degradation on this cross-modal transfer problem. The experiments provide an optimistic result which indicates cross-modal domain adaptation can serve as a low-cost Sim2Real approach.

## 2 RELATED WORK

To our best knowledge, this work is the first to address cross-modal domain adaptation in RL setting. We will discuss two research areas closely related to this topic, which are, (1) unsupervised visual domain adaptation in RL and (2) image-input representation learning in RL.

### 2.1 VISUAL DOMAIN ADAPTATION IN RL

Unsupervised visual domain adaptation aims to map the source domain and the target domain to an aligned distribution without pairing the data. Prior methods fall into two major categories: feature-level adaptation, where domain-invariant features are learnt (Gopalan et al., 2011; Caseiro et al., 2015; Long et al., 2015; Ganin et al., 2017), and pixel-level adaptation, where pixels from a source image used to generate an image that looks like one from the target domain (Bousmalis et al., 2017; Yoo et al., 2016; Taigman et al., 2017; Hoffman et al., 2018).

Pixel-level adaptation is challenging when data from two domains are unpaired. Prior works tackle this problem by using GANs (Goodfellow et al., 2014) conditioned on simulated images to generate

realistic images. Gamrian & Goldberg (2019) transfers policies from Atari Games (Bellemare et al., 2015) to modified variants by training a GAN to map images from the target domain to the source domain. GraspGAN (Bousmalis et al., 2018) addresses domain adaptation in robotic grasping by having the GAN reproduce the segmentation mask for the simulated image as an auxiliary task, including the robot arm, objects, and the bin. RCAN (James et al., 2019) adopts ideas from domain randomization by learning a mapping of images from randomized simulations to a canonical simulation and treating the real world just as one of the random simulations. RL-CycleGAN (Rao et al., 2020) unifies the learning of a CycleGAN (Zhu et al., 2017) and an RL policy, claiming better performance by learning features that are most crucial to the Q-function in RL.

Image-to-image domain adaptation can somewhat bypass the ill-posedness for distribution matching (See Fig. 1(a)) since it often enjoys an implicit advantage that images differ locally, in color, textile, lighting, but resembles globally between two domains, while images and states differ essentially. Some works impose extra structural constraints (e.g., segmentation, geometry) (Fu et al., 2019; Bousmalis et al., 2018), but such tricks fail in image-to-state domain adaptation either. In this work, we force the mapped states to follow transition consistency by using a recurrent structure (See Fig. 1(b)) and to be able to recover the pre-learned policy. We also employ a stochastic mapping function with the help of a variational encoder that is more robust to target domain data noise.

## 2.2 REPRESENTATION LEARNING IN RL

Representation learning aims to transform high-dimensional data into lower-dimensional vector representations, which suit RL better. It is widely accepted that learning policies from states (embeddings) is significantly more sample-efficient than learning from pixels, both empirically (Kaiser et al., 2020; Tenenbaum, 2018; Tassa et al., 2018) and theoretically (Jin et al., 2020).

Sequential auto-encoder is a common network structure to learn state representations by minimizing reconstruction loss. Early works on DRL from images (Ha & Schmidhuber, 2018; Lange et al., 2012; Lange & Riedmiller, 2010) use a two-step learning process where an auto-encoder is first trained to learn a low-dimensional representation, and subsequently a policy or model is learned based on this representation. Later works on model-based RL improve representation learning by jointly training the encoder and the dynamics model end-to-end (Watter et al., 2015) – this has been proved effective in learning useful task-oriented representations. PlaNet (Hafner et al., 2019) learns a hybrid of stochastic and deterministic latent state models using a reconstruction loss. SOLAR (Zhang et al., 2019) combines probabilistic graphic models with a simple network structure to fit local linear transitions. Some recent works adopt advancements in unsupervised representation learning. CURL (Laskin et al., 2020b) utilizes contrastive learning methods to capture essential information in an image that distinguishes from others, though later works (Laskin et al., 2020a; Kostrikov et al., 2020) point out that data augmentation may play the major role here.

Our work utilizes a sequential variational encoder structure to capture sequential information from trajectories. The main difference between our work and representation learning is whether the state space is predefined. We add extra supervised information to guide the training of the mapping by minimizing the distance between the distributions of the mapped states and the original states, and by enforcing the policy to recover the actions from the mapped states. As a result, we successfully learn states that match the ground-truth simulator states well. The mapped states can be directly fed into the pre-trained policy network.

## 3 CROSS-MODAL DOMAIN ADAPTATION WITH SEQUENTIAL STRUCTURE

Our work follows the problem setting similar to previous methods that tackle visual domain adaptation problems in RL. We have a policy $\pi$ pre-trained in the source domain (state) and a dataset pre-collected in the target domain (image). The task is to learn a mapping $q_\phi$ from images to states. In the deployment, agents interact with a new policy $\tilde{\pi}(o) = (\pi \circ q_\phi)(o)$, where $\circ$ denotes function composition. During the training of the mapping function, only the source domain is accessible.

This section is organized as follows. Sec. 3.1 formulates the cross-modal domain adaptation as a variational inference problem. Sec. 3.2 decomposes the variational inference problem into several feasible optimization objectives. Sec. 3.3 proposes a residual network structure to handle the complex long-horizon training of the sequential structure.

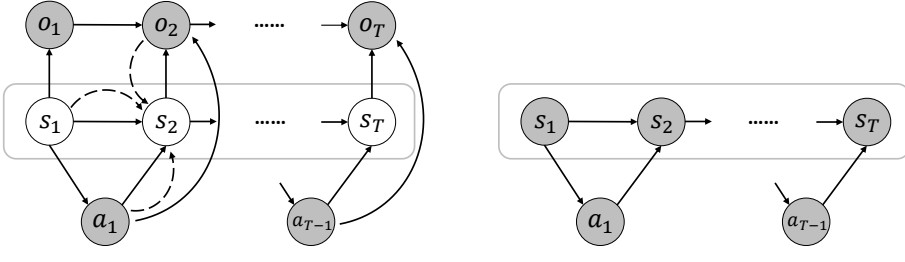

(a) generation process of real world      (b) generation process of simulation

Figure 2: Illustration of the generation in the real world and simulation domains respectively. All nodes are random variables. Shaded nodes are observable variables. The solid line denotes the generation process and dashed lines denote the inference process. There is an unpaired correspondence between two state trajectories surrounded by the rounded rectangle. Note that we include policy $\pi$ in both generation process, which corresponds to the edge from state $s_t$ to action $a_t$.

### 3.1 DOMAIN ADAPTATION AS VARIATIONAL INFERENCE

The goal of domain adaptation is to find a mapping from the target domain to the source domain, which is images to states in our case. We first model the generation process of the target domain and source domain respectively and the connection between them, as illustrated in Fig. 2. The initial state follows distribution $p(s_1)$. The transition function $\hat{p}_\varphi(s_t \mid s_{t-1}, a_{t-1})$, modeled as a feed-forward neural network with parameters $\varphi$, predicts the current state from the previous state and previous action. The decoder $p_\theta(o_t \mid s_t, a_{t-1}, o_{t-1})$, modeled as a deconvolution network with parameters $\theta$, reconstructs the current observation from the current state, previous observation and previous action. In practice, we model these distributions as multivariate Gaussian distributions. We allow $p_\theta(o_t \mid \cdot)$ dependent on $o_{t-1}$ so that some irrelevant patterns in images can be reconstructed in an auto-regressive manner. Such a conditional distribution is feasible in practice since we always have ground-truth $o_{0:t-1}$ at timestep $t$ in both training and deployment phases. A sequential generation process enjoys an extra benefit of solving Partially Observable Markov Decision Process (POMDP) since a single image can not reveal the full state of the environment in general.

The key point that distinguishes our method from conventional representation learning is the additional constraint that the mapped state trajectories should match the simulation state trajectories. See rounded rectangle parts in Fig. 2. Such a constraint may somewhat require that the underlying transition dynamics are the same in two domains. Introducing distributions can to some extent relax the assumption and increase robustness when there is a small mismatch between dynamics in source and target.

$$s_1 \sim p(s_1) \quad s_t \sim p(s_t \mid s_{t-1}, a_{t-1}) \quad o_t \sim p(o_t \mid s_t, a_{t-1}, o_{t-1}) \tag{1}$$

The overall optimization problem can be formulated as a variational inference problem described in Eq. 2, where we want to learn a posterior distribution to approximate the ground-truth distribution.

$$\min \mathbb{E}_{\tau^r} \left[ \mathcal{D}_{\mathrm{KL}} \left[ q_\phi(\tau^s \mid \tau^r) \mid\mid p(\tau^s \mid \tau^r) \right] \right] \tag{2}$$

where $\tau$ is a trajectory, $s$ and $r$ indicate whether the trajectory is from the source or target domain, $p(\cdot)$ is the ground-truth distribution, $q_\phi(\cdot)$ is the mapping function we want to learn, and $D_{KL}$ computes the Kullback-Leibler divergence. Modeling the optimization as a trajectory distribution matching can naturally handle the stochasticity of environments and policies, and the possible noise in data collected in the real world. The Evidence Lower Bound (ELBO) of this variational problem can be formulated as:

$$\max \mathbb{E}_{\tau^r} \left[ \mathbb{E}_{q_\phi(\tau^s \mid \tau^r)} \left[ \log p_\theta(\tau^r \mid \tau^s) \right] - \mathcal{D}_{\mathrm{KL}} \left[ q_\phi(\tau^s \mid \tau^r) \mid\mid p^\pi(\tau^s) \right] \right] \tag{3}$$

The derivation of ELBO follows a common practice using Jensen inequality. A detailed derivation can be found in Appendix A. The first term maximizes the reconstruction probability, in order to enforce that the mapped states $\hat{s}$ can recover both observations $o$ and action $a$. The second term

---

**Algorithm 1** Training Procedure of CODAS

---

**Input:** Simulator with oracle dynamics $p$; Policy $\pi(a \mid s)$ pre-trained in the simulated dynamics
$p(s' \mid s, a)$; Target-domain Trajectory Dataset of images $\mathcal{D}^r = \{(o_0^r, a_0^r, ..., o_T^r)_i\}$; Number of
iteration $N$.

**Output:** Mapping function $f : \mathcal{O} \to \mathcal{S}$.

Initialize the sequential mapping function $f(s_t \mid s_{t-1}, a_{t-1}, o_t^r)$.

   **for** $n = 1$ to N **do**

      Sample a batch of target domain trajectories $\tau^r = \{(o_0^r, a_0^r, ..., o_T^r)_i\}$ from $\mathcal{D}^r$

      Initialize RNN with a zero state.

      Infer the corresponding state trajectories $\tau^s = \{(\hat{s}_1, ..., \hat{s}_T)_i\}$ via the mapping function $f$;

      Rollout one step with the oracle simulation dynamics $p(s' \mid s, a)$ for each state-action pair
         in $\tau^s$ to construct the transition dataset $\mathcal{D}_{\hat{s}} = \{(\hat{s}_i, a_i, s_{i+1})\}$;

      **for** $d = 1$ to D **do**

         Update the discriminator $D$ by maximizing Eq. 13;

      **end for**

      Update the mapping function (details are in Algorithm 2);

   **end for**

---

enforces the alignment of the distributions of the mapped trajectories and the trajectories collected
in the simulator. The policy $\pi$ in source domains included in the second term does introduce a new
assumption that target domain data are collected by a known behavioral policy. This assumption
is mild and is implicitly or explicitly introduced in previous works (Kim et al., 2020; Gamrian &
Goldberg, 2019).

## 3.2 DIFFERENTIABLE OPTIMIZATION OBJECTIVES

The ELBO defined in Eq. 3 contains terms involving distributions over the entire trajectory, and thus
is impractical to solve directly. Given the generation process defined in the previous section, we can
decompose the joint probability into the multiplication of one-step probabilities. Eq. 4 shows the
result after the decomposition. For brevity, we will use $\hat{s}, \hat{o}$ to denote $s, o$ outputted by networks and
omit the networks themselves.

$$\max \mathbb{E}_{\tau^r} \left[ \sum_{t=1}^{T} \mathbb{E}_{q_\phi(\hat{s}_t \mid o_t, \hat{s}_{t-1}, a_{t-1})} [\log p_{\theta_s}(\hat{o}_t^r \mid o_{t-1}, \hat{s}_t, a_{t-1}) + \log p_{\theta_\pi}(a_{t-1} \mid s_{t-1})] \right.$$
$$\left. - \mathcal{D}_{\mathrm{KL}}[q_\phi^\pi(\tau_s \mid \tau_r) \mid p^\pi(\tau_s)] \right. \tag{4}$$

A direct computation of the second $\mathcal{D}_{\mathrm{KL}}$ term is intractable. Following the idea that the optimiza-
tion process of GAN is equivalent to minimizing a certain distance measure between two distri-
butions (Nowozin et al., 2016), we can use the optimization objective of GAN as the surrogate
objective of minimizing the $\mathcal{D}_{\mathrm{KL}}[q_\phi(\tau^s \mid \tau^r) \| p^\pi(\tau^s)]$. For implementation simplicity and train-
ing stability, we still choose to use the original GAN optimization objective, which is equivalent to
minimizing JS divergence. The optimization objective is formulated in Eq. 5.

$$\ell_{\mathrm{D}}(\theta, \omega) = \mathbb{E}_{s \sim D^s}[D_\omega(s, a)] + \mathbb{E}_{\tau^r \sim D^r}[\log(1 - D_\omega(\hat{s}, a))] \tag{5}$$

Here, the discriminator takes state, rather than trajectory, as input for better practicality, where "real"
samples are simulation states and "fake" samples are mapped states. The latter term is the objective
of the generator - $q_\phi$ in our case. The sequential structure is preserved in the generator. This part is
different from previous works like VAE-GAN (Larsen et al., 2016) and Causal InfoGAN (Kurutach
et al., 2018), that discriminate against the reconstructed output. Instead, we discriminate against the
states after the variational encoder.

The reconstruction of action $a$ can be optimized in an end-to-end manner. However, if a differ-
entiable $\pi$ of simulation states is available (which is the usual case), we can replace $p_{\theta_\pi}$ with $\pi$,
resulting an objective in Eq. 6. A fixed $\pi$ here can guide $p_{\theta_s}$ to output states that yields a similar $a$.
Previous works (Schrittwieser et al., 2019) also discovered that recovering $a$ is helpful in improving
the representation learning in RL.

---

**Algorithm 2** Detailed Training Procedure of Mapping Function

---

**Input:** Simulator with oracle dynamics $p$; Policy $\pi(a \mid s)$ pre-trained in the simulation dynamics $p(s' \mid s, a)$;
   # Initialization.
   Pre-train DM $t_\varphi$ with $\mathcal{D}^s = \{(s_i, a_i, s_{i+1})\}$.
   # Update per iteration.
   Update the the DM $t_\varphi$ $\hat{p}$ by minimizing Eq. 8 using $\mathcal{D}^{\hat{s}}$ and $\mathcal{D}^s$;
   Copy the new parameters $\varphi$ of DM $t_\varphi$ to the embed-DM in $q_\phi$;
   **for** $m = 1$ to M **do**
      Update the mapping function $q_\phi$ by minimizing Eq. 7;
      Update the reconstruction function $p_\theta$ by minimizing the first term in Eq. 7;
   **end for**

---

$$\ell_{\text{policy}} = \mathbb{E}_{(\tau^r \sim D^r}[\sum_{t=1}^{T} [\log \pi(a_t^r \mid \hat{s}_t))] \tag{6}$$

Combining all the aforementioned losses together, i.e, reconstruction loss, policy loss and generation loss, the complete optimization objective of the mapping function is as follows:

$$\ell_{\text{mapping}} = \mathbb{E}_{\tau^r \sim D^r} \sum_{t=1}^{T} \log p(\hat{o}_t \mid o_t) + \lambda_D \log(1 - D_\omega(\hat{s}_t)) + \lambda_\pi \log \pi(a_t^r \mid \hat{s}_t)) \tag{7}$$

where $\lambda_D$ and $\lambda_\pi$ are hyper-parameters. Both the decoder and the policy are fixed during the training process of the mapping function. The loss function for $p_\theta$ is just the first term in Eq. 7. Similarly, the mapping function $q_\phi$ is fixed during the training process of the reconstruction function $p_\theta$.

### 3.3 State Inference Model with Embedded Dynamics

Since a trajectory of practical RL problems often lasts hundreds or thousands timesteps, training RNN on such a long-horizon trajectory is difficult. The previous success of ResNet (He et al., 2016) shows that a residual structure can simplify the learning target by predicting small residuals, resulting in a remarkable performance increase in learning ultra-deep neural networks.

Adopting a similar idea, we incorporate a residual structure in the RNN to help stabilize the training process. We have a Dynamics Model (*DM*) trained independently using transition tuples collected in the simulator. This structure further forces the mapped states to follow the simulation dynamics. Its parameters are periodically updated to *embed-DM*, in order to provide an "average" estimation of the next states $\bar{s}_t$ (See Fig. 3). The job of the rest part, instead, is simplified to just output a correction. With the proposed structure model, the mapped states follow the transition dynamics in the simulator better.

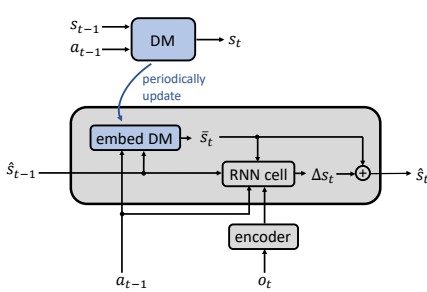

Figure 3: Model structure with embed-DM

*DM* is first trained using batches of transition tuples collected in the simulator. During the training process of the mapping function, the dynamics model is trained online using $D^{\hat{s}} = \{(\hat{s}_t, a_t, s_{t+1} \sim p(s'|\hat{s}_t, a_t)\}$ and is periodically updated to the mapping function. That is, we reset the simulator to the mapped states $\hat{s}_t$ and then rollout with a one-step oracle simulation transition to get $s_{t+1}$. The optimization objective of the dynamics model is an MSE loss in Eq. 8. Algorithm 2 demonstrates a detailed training and updating procedure of the dynamics model and the entire mapping function.

$$\ell_{\text{dynamics}} = \mathbb{E}_{(s,a,s') \sim D^s \cup D^{s'}} [(t_\varphi(s, a) - s')^2] \tag{8}$$

## 4 EXPERIMENTS

We evaluate our method in 6 MuJoCo environments from OpenAI Gym, namely InvertedPendulum, InvertedDoublePendulum, HalfCheetah, Hopper, Swimmer and Walker2d. We define the rendered images as the target domain, and the original observations as the source domain. The pre-collected dataset in the target domain contains 600 trajectories and is collected by a sub-optimal policy. Please refer to Appendix C for a detailed experiment setting.

We modify state-of-the-art methods in same-modal domain adaptation for comparison, namely GAN and CycleGAN. GAN uses the same model structure of the encoding network in CODAS, and is trained on pre-collected state and image datasets. The state dataset is of the same size as the image dataset. As described in Sec. D.6, CycleGAN fails in all environments. We also compare with Behavioral Cloning (BC) considering target domain data are collected by a (sub)optimal policy. BC trains a policy in a supervised manner using $(o_t^r, a_t^r) \sim D^r$. To mitigate the partial observability issue mentioned in Sec. 3, we stack every 4 consecutive images as the new input to the BC algorithm. All methods are trained until convergence. Implementation details for all methods can be found in Sec. B. Due to some numerical instabilities[1] inside MuJoCo, we cannot get the oracle transition function in HalfCheetah and InvertedDoublePendulum. CODAS in these two environments are implemented without embed-DM.

Experiment results in this section will answer the following three questions:

1) Does CODAS enable polices to transfer from states to images? Does CODAS outperform state-of-the-art methods?

2) Does every component in CODAS contribute to the overall performance?

3) Is CODAS robust to small mismatches in environment dynamics between the source and target domains?

To focus on the performance of the adaptation process, we use reward ratio as the metric, which is defined as $r_{ratio} = \frac{r}{r^*}$, where $r$ and $r^*$ are the cumulative return of the adapted policy and the optimal policy trained on states respectively. The quantitative performance of the optimal policy trained on states and rendered images using PPO is given in Sec. D.2.

### 4.1 PERFORMANCE

Training curves of all methods are shown in Fig. 5, x-axis being training iterations, y-axis being the reward ratio. Each iteration, every method is updated using a batch of 20 trajectories. CODAS reaches an average of 70% reward ratio after adaptation, providing an optimistic result on future applications of cross-modal domain adaptation. BC performs well in simple environments (HalfCheetah and Inverted Pendulum), but poorly in rest environments, especially in those with an early termination. GAN performs even worse than BC in most environments, which suggests

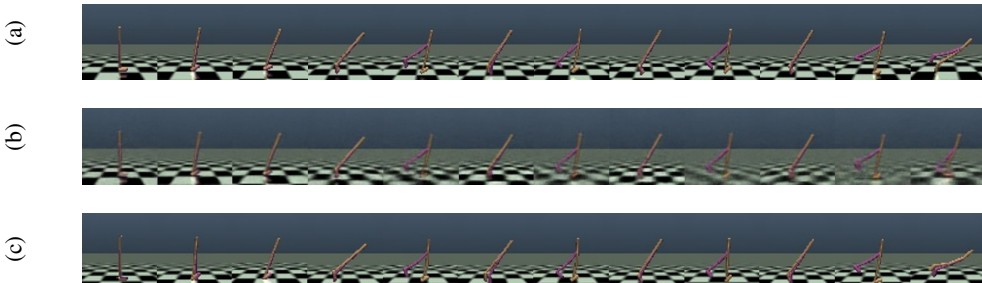

Figure 4: A visual illustration of (a) original images (b) reconstructed images (c) re-rendered images by setting the simulator to mapped states.

---

[1]MuJoCo engine outputs different $s'$ given exact same $(s, a)$ as input due to its inner inaccessible random states.

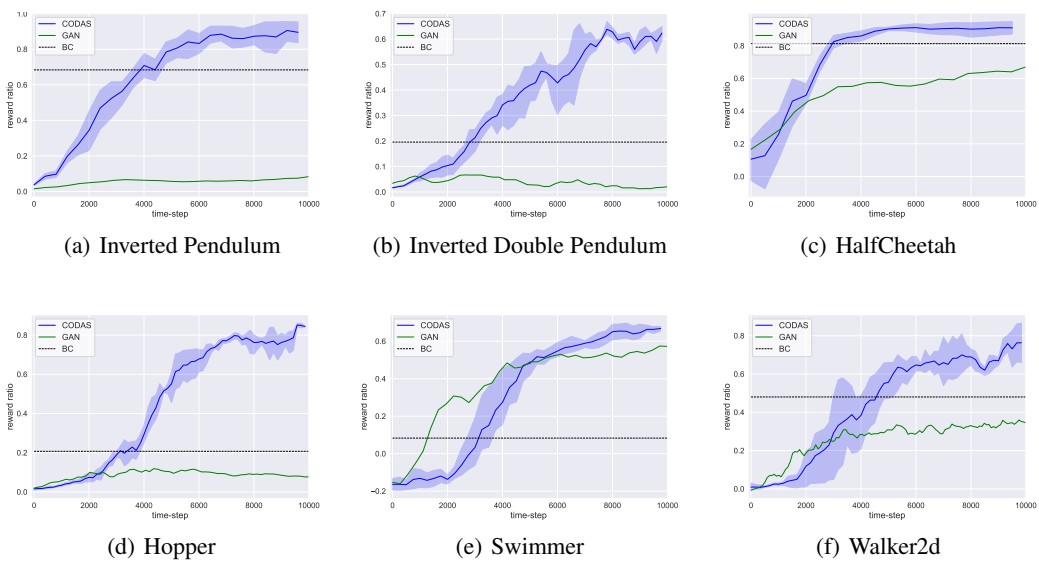

Figure 5: Training curves of different methods on cross-modal domain adaptation

mapping images to a predefined interpretable and meaningful state space is more difficult than end-to-end policy learning.

A visual illustration of the mapped states accuracy is shown in Fig. 4. Both reconstructed images and re-rendered images match the original ones well. It is worth noting that re-rendered images can match the last falling frames, which are sparse in the dataset, well. Quantitative results on mapping states errors of GAN and our methods can be found in Sec. D.3.

## 4.2 ABLATION STUDIES

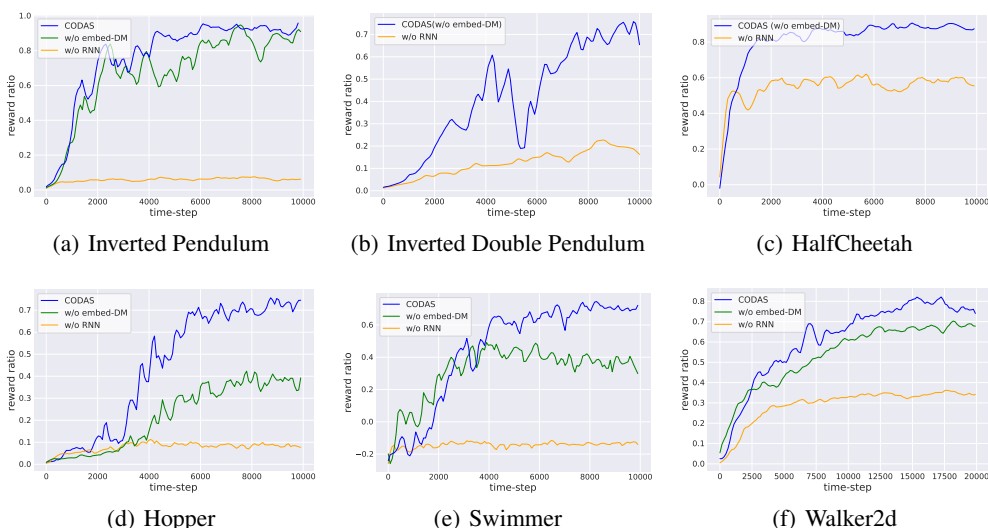

Figure 6: Ablation studies by removing embed-DM and RNN in CODAS

Since CODAS contains multiple ideas, we conduct additional experiments to understand the contribution of each component to the overall performance. We conduct ablation studies to demonstrate (1) if sequential structure matters in the cross-modal domain adaptation in RL and (2) if embed-DM

improves the long-horizon inference. Since the embed-DM is based on sequential structure, we only test on two ablated variants, i.e., CODAS w/o embed-DM and w/o both embed-DM and RNN.

A comparison of all ablated variants is shown in Fig. 6. Both axes are of the same meanings as above. In all the environments, mapping functions with sequential structure yield better deployment performance. In four environments where embed-DM is applicable, embed-DM helps further improve the performance. As we expect, the major purpose of incorporating embed-DM is to simplify the learning process. If RNN itself can learn a reasonable mapping, the improvement of embed-DM will be marginal (Walker2d). The learning process is also a little unstable in InvertedDoublePendulum, where embed-DM is not available.

### 4.3 ROBUSTNESS TO DYNAMICS MISMATCH

To analyze the CODAS's robustness to dynamics mismatch, we modify the dynamics in the source domain. Specifically, we reconfigure the friction coefficient in Hopper. The result can be found in Figure 11. The results in Hopper both with 110% and 120% friction still demonstrate a robust mapping. Besides, the mismatch of dynamics even not slow down the speed of learning. The result suggests that the CODAS technique has the potential to be deployed in more realistic applications where the dynamics in two domains can not match exactly. We leave this analysis in future works.

## 5 CONCLUSION

In this work, we propose Cross-Modal Domain Adaptation with Sequential structure (CODAS). CODAS enables a new paradigm of Sim2Real - adapting policies trained on simulated states to real-world with image inputs. We believe this setting is valuable in real-world applications of RL by exempting us from the tedious work of building and running rendering engines. Previous methods that use GANs fail on this problem since global structural resemblance between two same modal domains no longer exists, while our method succeeds by fully leveraging the sequential structure and other auxiliary information provided by the policy and dynamics underlying RL problem. We first model the cross-modal domain adaptation problem as a variational inference problem and decompose it into several feasible optimization objectives. To better solve the complex long-horizon sequential mapping problem, we propose a residual network structure. We validate the proposed method by adapting policies from images to states on various MuJoCo environments. Our results provide an optimistic results of cross-modal domain adaptation as a low-cost Sim2Real approach.

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

## A   DERIVATION OF OPTIMIZATION OBJECTIVES

Assume the distribution of trajectories in the simulator $p^\pi(\tau_s)$ is calculated from the real world trajectories $\tau_r$. We want to match $q_\phi^\pi(\tau_s|\tau_r)$ to the ground-truth distribution $p_\theta^\pi(\tau_s|\tau_r)$, leading to the optimization objective:

$$\min \mathbb{E}_{\tau^r}[\mathcal{D}_{\mathrm{KL}}[q_\phi^\pi(\tau_s|\tau_r)||p_\theta^\pi(\tau_s|\tau_r)]] \tag{9}$$

To maximize the objective, we first transform it into the Evidence Lower Bound (ELBO):

$$\mathcal{D}_{\mathrm{KL}}[q_\phi^\pi(\tau_s|\tau_r)||p_\theta^\pi(\tau_s|\tau_r)]$$
$$=\mathbb{E}_{q_\phi^\pi(\tau_s|\tau_r)}[\log \frac{q_\phi^\pi(\tau_s|\tau_r)}{p_\theta^\pi(\tau_s|\tau_r)}]$$
$$=\mathbb{E}_{q_\phi^\pi(\tau_s|\tau_r)}[\log q_\phi^\pi(\tau_s|\tau_r) - \log \frac{p_\theta^\pi(\tau_s,\tau_r)}{p^\pi(\tau_r)}]$$
$$=\mathbb{E}_{q_\phi^\pi(\tau_s|\tau_r)}[\log q_\phi^\pi(\tau_s|\tau_r) - \log p_\theta^\pi(\tau_r,\tau_s)] + \log p_{(}^\pi\tau_r)$$

The last term $p_\theta^\pi(\tau_r)$ is a constant with regards to the parameter $\theta$, and thus can be ignored in the optimization process. The minimization can be reduced to $\min .\mathbb{E}_{\tau^r}[\mathbb{E}_{q_\phi^\pi(\tau_s|\tau_r)}[\log q_\phi^\pi(\tau_s|\tau_r) - \log p_\theta^\pi(\tau_r,\tau_s)]]$. Since the sampled trajectories can be considered as an i.i.d., the first term of expectation can be further re-written as

$$\mathbb{E}_{q_\phi^\pi(\tau_s|\tau_r)}[\log q_\phi^\pi(\tau_s|\tau_r) - \log p_\theta^\pi(\tau_r,\tau_s)]$$
$$=\mathbb{E}_{q_\phi^\pi(\tau_s|\tau_r)}[\log \frac{q_\phi^\pi(\tau_s|\tau_r)}{(p_\theta^\pi(\tau_r|\tau_s)p_\theta^\pi(\tau_s))}] \tag{10}$$
$$= -\mathbb{E}_{q_\phi^\pi(\tau_s|\tau_r)}[\log p_\theta^\pi(\tau_r|\tau_s) - \mathcal{D}_{\mathrm{KL}}[q_\phi^\pi(\tau_s|\tau_r) \mid\mid p^\pi(\tau_s)]$$

Based on the generation process, we can decompose the first term based on trajectories to multiplication of terms based on states.

$$q_\phi^\pi(\tau^s \mid \tau^r) = \prod_{t=1}^{T_\tau} q_\phi(s_t \mid s_{t-1}, o_t, a_{t-1}^r), \quad o_t, a_{t-1} \sim \tau^r$$
$$p_\theta^\pi(\tau^r \mid \tau^s) = \prod_{t=1}^{T_\tau} p_\theta(o_t \mid o_{t-1}, s_t, a_{t-1}), \quad s_t, a_{t-1} \sim \tau^s \tag{11}$$

$$\mathbb{E}_{\tau^r}[\mathbb{E}_{q_\phi^\pi(\tau_s|\tau_r)}[\log p_\theta^\pi(\tau_r|\tau_s)]$$
$$=\mathbb{E}_{\tau^r}[\mathbb{E}_{q_\phi^\pi(\tau_s|\tau_r)}[\sum_{t=1}^{T} \log p_\theta(\hat{o}_t|o_{t-1}, \hat{s}_t, a_{t-1})] \tag{12}$$
$$=\mathbb{E}_{\tau^r}[\sum_{t=1}^{T}(\mathbb{E}_{q_\phi(\hat{s}_t|o_t, \hat{s}_{t-1}, a_{t-1})}[\log p_\theta(\hat{o}_t|o_{i-1}, \hat{s}_t, a_{i-1})]]$$

A direct computation of the second term $\mathcal{D}_{\mathrm{KL}}$ is intractable. Following the idea that the optimization process of GAN is equivalent to minimizing a certain distance measure between two distributions (Nowozin et al., 2016), we can use the optimization objective of GAN as the surrogate objective of minimizing the $\mathcal{D}_{\mathrm{KL}}[q_\phi(\tau^s \mid \tau^r) \mid\mid p^\pi(\tau^s)]$ by introducing a new discriminator $D_\omega$ parameterized by $\omega$ that maximizes Eq. 13,

$$\ell_{\mathrm{D}}(\theta,\omega) = \mathbb{E}_{s\sim D_s}[1 + \log \frac{D_\omega(s)}{1-D_\omega(s)}] + \mathbb{E}_{\tau^r\sim D^r}[\frac{D_\omega(\hat{s})}{1-D_\omega(\hat{s})}] \tag{13}$$

For implementation simplicity and training stability, $(\frac{D_\omega(s)}{1-D_\omega(s)})$ can become arbitrarily large, we still choose to optimize the original GAN optimization objective, which is formulated as Eq. 14.

$$\ell_{\mathrm{D}}(\theta, \omega) = \mathbb{E}_{s \sim D_s}[D_\omega(s)] + \mathbb{E}_{s \sim f_\theta(o^r)}[\log(1 - D_\omega(s))] \tag{14}$$

## B  IMPLEMENTATION DETAILS

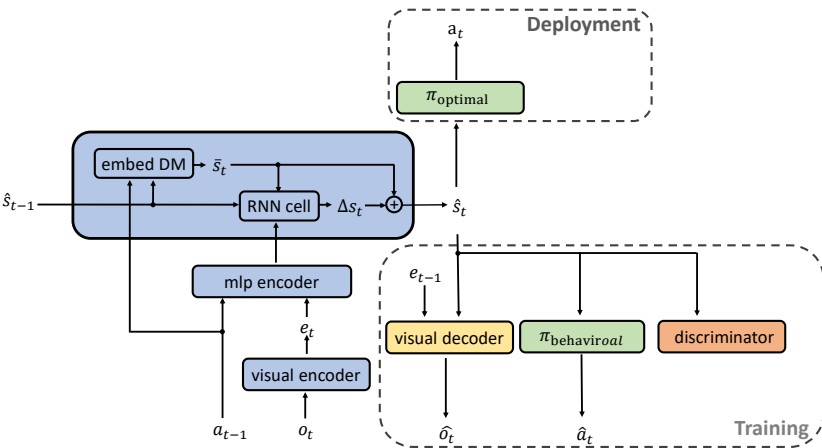

Figure 7: Illustration of full network structure. Blue parts denote mapping function $q_\phi$; Yellow and green parts denote reconstruction function $p_\theta$. Green part is always fixed in the entire training process.

Fig. 7 gives an illustration of the overall structure of our method. Both $\pi_{\mathrm{optimal}}$ and $\pi_{\mathrm{behavioral}}$ are polices of the source domain, where $\pi_{\mathrm{optimal}}$ is the optimal policy trained in the source domain and $\pi_{\mathrm{behavioral}}$ is a policy policy in the source domain that mimics the behavior of the dataset collection policy (in terms of the cumulative returns). In rare cases when a differentiable $\pi_{\mathrm{behavioral}}$ is not available (e.g. the policy is a decision tree), it can be replaced by a trainable network here. Detailed network structure and hyperparameters are listed in Sec. B.1, all of which are the same for every environment. To help balance the influence of the residual $\Delta s_t$, we introduce a new hyperparameter $\lambda_{\mathrm{res}}$, s.t,

$$\hat{s}_t = \bar{s}_t + \lambda_{\mathrm{res}}\Delta s_t \tag{15}$$

### B.1  HYPERPARAMETERS AND NETWORK STRUCTURE

Table 1: Hyperparameters for the Reconstruction Network

| Name | Value |
|---|---|
| **General** | |
| $\lambda_D$ | 50.0 |
| $\lambda_\pi$ | 1.0 |
| $\lambda_{\mathrm{res}}$ | 1.5 |
| batch size | 20 (trajs) |
| mapping update iter $M$ | 5 |
| discriminator update iter $D$ | 1 |
| DM update iter | 10 |
| DM batch size | 512 |
| DM sync freq | 1 |
| grad clip norm | 10 |

Table 2: Hyperparameters for the Reconstruction Network (Cont.d)

| Name | Value |
|---|---|
| **RNN** | |
| RNN type | GRU |
| RNN hidden size | $128,\ 128$ |
| **Dynamics Model** | |
| Hidden sizes | $512,\ 512,\ 512$ |
| Activation function | tanh |
| Learning rate | $1 \times 10^{-4}$ |
| **Policy** | |
| Hidden sizes | $64, 64$ |
| Activation function | tanh |
| **Discriminator** | |
| Hidden sizes | $256,\ 256,\ 256,\ 256,\ 256$ |
| Activation function | relu |
| Layer norm | true |
| Learning rate | $5 \times 10^{-5}$ |
| **Visual Encoder** | |
| Hidden sizes | Conv (4, 4, 32), Conv (4, 4, 64), , Conv (4, 4, 128), Conv (4, 4, 256), 256, 256, 256 |
| Activation function | relu |
| Layer norm | true |
| Learning rate | $5 \times 10^{-5}$ |
| **Visual Decoder** | |
| Hidden sizes | 256, 256, 1024, Deconv (4, 4, 128), Deconv (4, 4, 64), , Deconv (4, 4, 32), Conv (4, 4, 3), |
| Activation function | relu |
| Layer norm | true |
| Learning rate | $5 \times 10^{-5}$ |
| **MLP Encoder** | |
| Hidden sizes | 256, 256 |
| Activation function | relu |
| Layer norm | true |
| Learning rate | $5 \times 10^{-5}$ |

## C    EXPERIMENT SETTING

We evaluate our method in 6 MuJoCo environments from OpenAI Gym, namely InvertedPendulum, InvertedDoublePendulum, HalfCheetah, Hopper, FixedSwimmer and Walker2d. FixedSwimmer is based on modifications proposed in previous works (Wang et al., 2019) to avoid sub-optimal policies by changing one sensor position. We treat the simulation observations as low dimensional simulation states $s$ and rendered images as real images.

Images are collected by "track camera" in HalfCheetah, Hopper, FixedSwimmer and Walker2d and "default camera" in InvertedPendulum and InvertedDoublePendulum. All images are resized to $[64, 64, 3]$ without any further pre-processing techniques in all environments. Examples of rendered images are shown in Fig. 8.

Two polices are independently trained until convergence using PPO (Schulman et al., 2017) for every environment. One of them is regarded as $\pi_{\text{target}}$ to collect the "real" image dataset. $\pi_{\text{target}}$ is a stochastic policy to mimic the real data collection process. The other is regarded as a pre-trained simulator policy $\pi_{\text{source}}$. The image dataset contains 600 episodes, each being truncated to a maximum length of 500. The evaluation of all methods are done based on this truncated dataset.

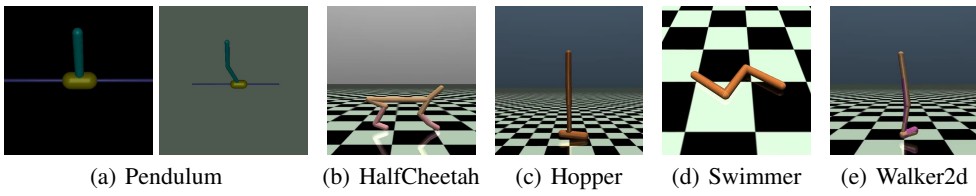

 (a) Pendulum    (b) HalfCheetah   (c) Hopper   (d) Swimmer   (e) Walker2d

Figure 8: Examples of rendered images of MuJoCo environments

# D EXTRA EXPERIMENT RESULTS

## D.1 AVERAGE REWARD RATIO OF ALL ENVIRONMENTS

| Method | CODAS(ours) | GAN | BC(max) | BC(final) |
|---|---|---|---|---|
| Reward Ratio | **70.1%** | 42.8% | 47.83% | 34.91% |

## D.2 PERFORMANCE OF THE OPTIMAL POLICY

Table. 3 shows the mean value and standard deviation of the un-discounted cumulative return of 100 trajectories collected by the optimal policy trained on states using PPO. The maximum episode length is set to 1000. Full training curves of policies trained on state space are provided in Fig. 9. The performance of policies trained on state space matches the public benchmarking results.[2]

Fig. 9 also provides training curves of the policy trained on images. We modify the network structure of actor and critic to adapt PPO to image input. In all environments, the policy perform poorly. The final performance of image input is calculated at $3.0 \times 10^6$ time-step, when the value loss has converged. As far as we know, there is no public performance benchmark of optimal policy trained on MuJoCo images. Some previous results on Deepmind Control Suite shows a better result than ours, partly due to a much clearer robots and background and tuned hyperparams.

Table 3: Performance of Optimal Policy on State Space

| Environment | Return | Environment | Return |
|---|---|---|---|
| Hopper | $2097 \pm 411$ | Swimmer | $325 \pm 5$ |
| Walker2d | $3669 \pm 587$ | InvertedPendulum | $775.5 \pm 319$ |
| HalfCheetah | $1580 \pm 35$ | InvertedDouble | $5201 \pm 1029$ |

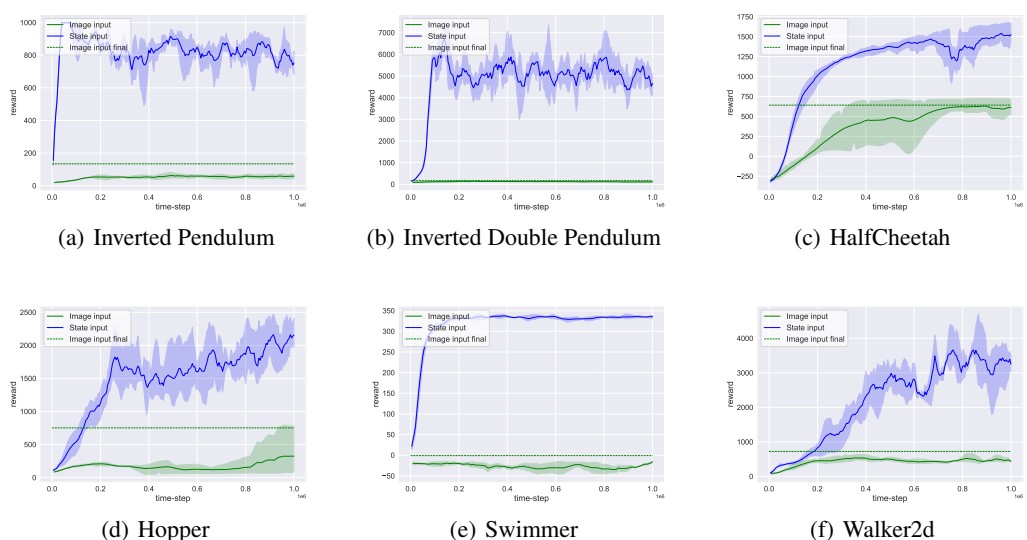

(a) Inverted Pendulum     (b) Inverted Double Pendulum     (c) HalfCheetah

(d) Hopper     (e) Swimmer     (f) Walker2d

Figure 9: Training curves of policy on state space and image space.

---

[2] https://spinningup.openai.com/en/latest/spinningup/bench.html

### D.3 QUANTITATIVE RESULTS OF STATE MAPPING ERROR

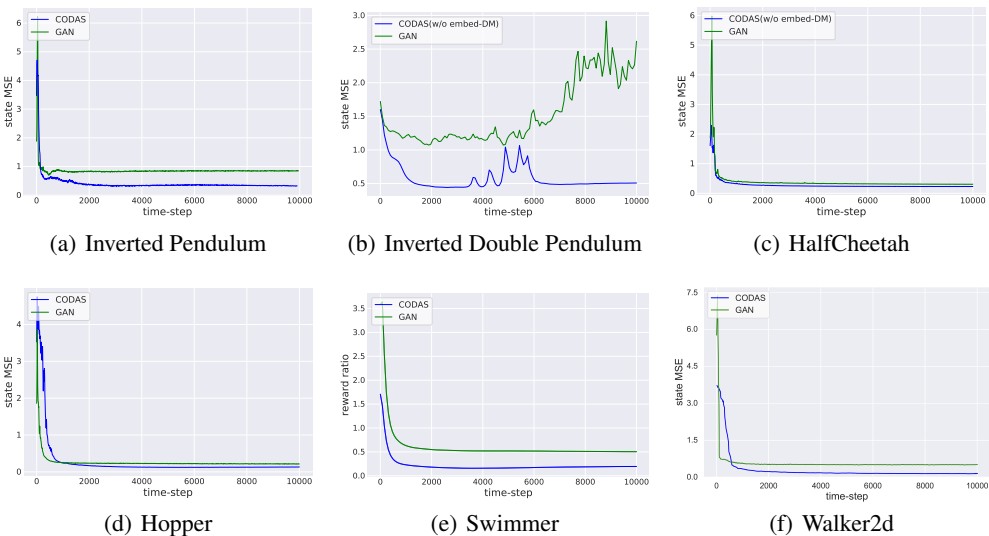

(a) Inverted Pendulum     (b) Inverted Double Pendulum     (c) HalfCheetah

(d) Hopper     (e) Swimmer     (f) Walker2d

### D.4 TRAINING CURVES OF BEHAVIORAL CLONING

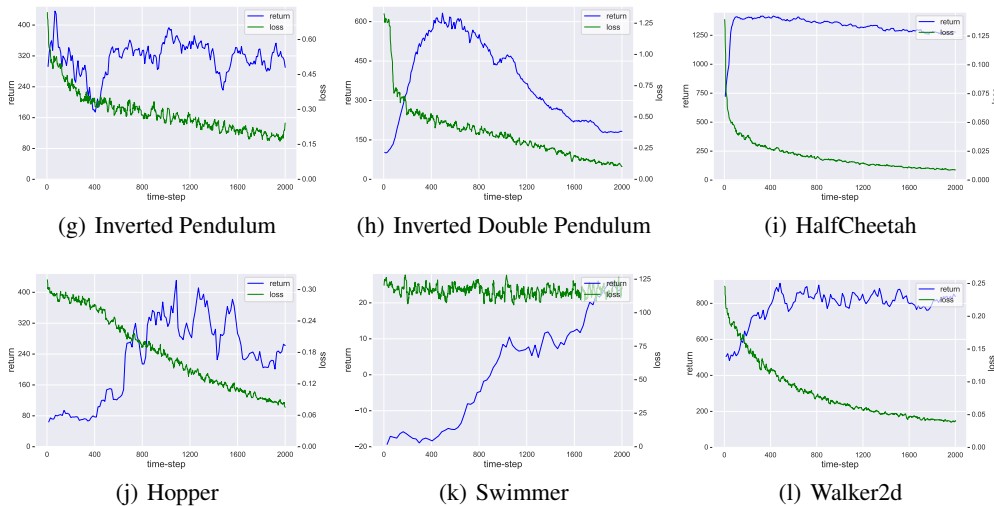

(g) Inverted Pendulum     (h) Inverted Double Pendulum     (i) HalfCheetah

(j) Hopper     (k) Swimmer     (l) Walker2d

Figure 10: Training curves of Behavior Cloning.

### D.5 ROBUSTNESS TO DYNAMICS MISMATCH

To test the robustness of CODAS, we manually change the friction of the target environment to create dynamics mismatches. Fig. 11 shows the reward ratio of CODAS in Hopper environment with different magnitudes of friction. The performance of CODAS remains stable even when the amplitude of friction reaches 20%, proving that CODAS is robust to mild dynamics mismatch. It is worth noting that the policy is trained without any technique such as domain randomization that improves robustness.

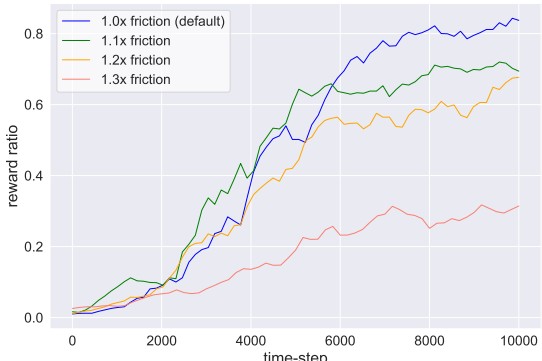

Figure 11: Reward ratio in Hopper with small dynamics mismatches.

### D.6 TRAINING CURVES OF CYCLEGAN

We planned to use CycleGAN, one of the state-of-the-art methods in domain adaptation, for comparison. Concretely, we remove the identity loss and use the network structure as ours (See Visual Encoder/Visual Decoder in Sec. B) as the generator network structure. However, the changes in the loss function and network structure may require a complete hyperparameter setting from image-to-image translation. We fail to get a workable CycleGAN in most environments. The best results so far are shown in Fig. 12. The training of state-to-image GAN in CycleGAN is not stable and may lead to the overall bad performance. It could be of the following reasons:

- id loss in the original CycleGAN to improve training stability is not applicable;
- network structures (e.g. UNet) that are specially designed for images generation are not applicable;

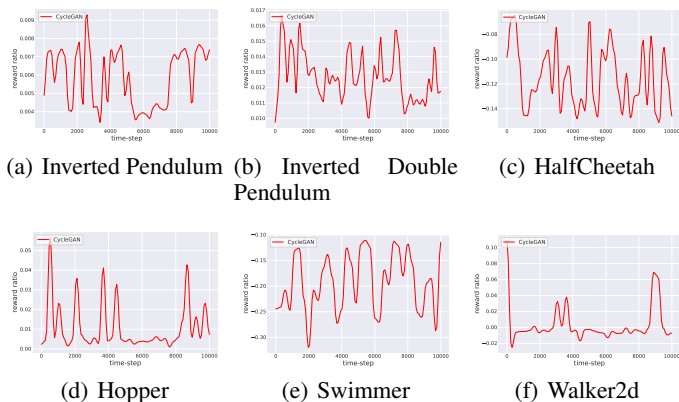

(a) Inverted Pendulum   (b) Inverted Double Pendulum   (c) HalfCheetah

(d) Hopper   (e) Swimmer   (f) Walker2d

Figure 12: Training curves of CycleGAN.

### D.7 GAN WITH ACTION LOSS

Fig. 13 shows the training curve of GAN with additional policy loss defined in Eq.6. Surprisingly, adding a fixed policy network does not help the learning process of naive GAN. It may be because the gradient from this fixed policy in the initial training stage is useless or even harmful to the training of the mapping function. The performance of GAN with policy loss drops significantly partly due to this loss is large in the modified Swimmer environment (can be found in the BC loss in Fig. 10 as well).

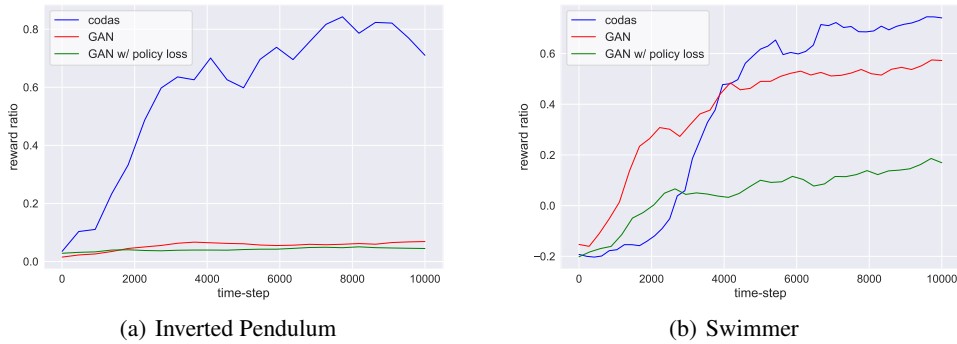

|  (a) Inverted Pendulum | (b) Swimmer |

Figure 13: Training curves of GAN with action loss.

## E A SIMPLE EXAMPLE OF FAILED DISTRIBUTION MATCHING

Fig. 14 shows an example of wrong mapping using GAN. Fig. 14(a) and Fig. 14(b) are two Gaussian Mixture Models of 4 modes with a linear transformation between them. The translucent circles denote the ground-truth correspondence between every mode in two domains and the dots denote the actual correspondence learned by GAN. Although GAN successfully matches the probability densities, it fails to map the target instances to their corresponding correct source instances. Under such a wrong mapping function, a neural network trained on samples in the source domain cannot be transferred to the target domain directly.

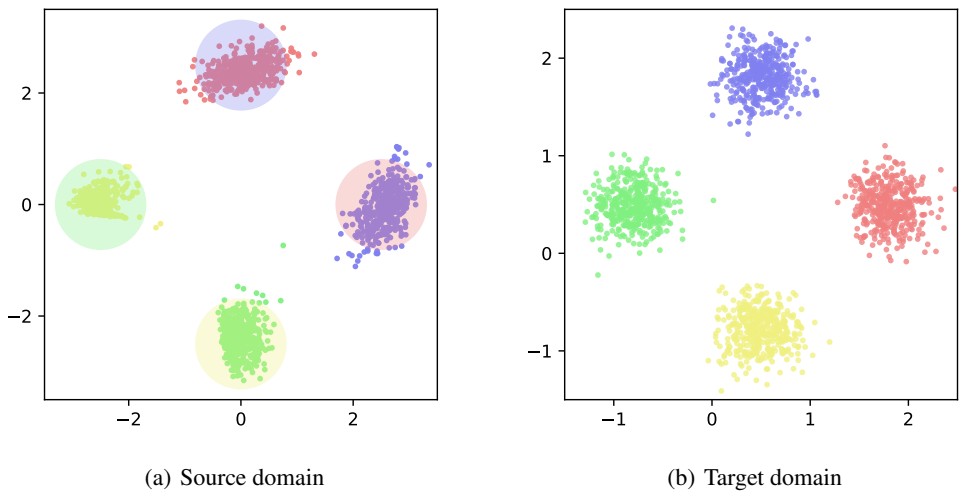

|  (a) Source domain | (b) Target domain |

Figure 14: An example of failed distribution matching under a linear transformation.

