# OpenReview forum: "Cross-Modal Domain Adaptation for Reinforcement Learning"
_ICLR.cc/2021/Conference — Reject_

### Official Review · AnonReviewer2 · 2020-10-26
**Good paper, but further details required**

**Rating:** 5
**Confidence:** 4

**Review:**

## Summary
The paper proposes a new approach for performing cross-modal domain adaptation, i.e. adapting a policy trained with inputs from modality A (eg low-dimensional environment state) to work with inputs from domain B (eg images). The main use case demonstrated in the paper is the adaptation of policies trained on states in a simulator to work on image inputs, which can be useful for eg real world deployment where states might not be available. While it is a very classic approach to separately train a perception module images --> state (eg in robotics), the main novelty of the presented method is, that this mapping can be learned without the need for paired [image, state] data.

## Strengths
- the discussed problem of cross-modal domain adaptation is very relevant, since it can allow for the transfer of policies from training in simulation to deployment in real environments
- being able to learn the image --> state mapping without paired data can be impactful since it is often tedious / impossible to get exact state annotations for real world data
- the proposed technical approach is novel to the best of my knowledge
- the method is evaluated on multiple environments and some of the design decisions are ablated + (preliminary) experiments on the robustness of the method are conducted

## Weaknesses
- **baselines are not described in detail**: the experimental section only mentions that "We modify state-of-the-art methods in same-modal domain adaptation for comparison" -- it is important to describe which method for same-modal adaptation was used and how it was modified for the cross-modal case to properly judge the results.
- **no investigation into why baseline does not work**: figure 1 provides one possible explanation (because of not taking dynamics into account), but later the paper mentions it could be because the necessary biases present in image-to-image translation are not present in image-to-state translation, finally it could be because the method was not tuned sufficiently. It would be good to show a more detailed investigation of this, particularly since the paper claims to be the first trying to apply domain adaptation techniques to the cross-modal case
- **only tested on visually clean environments**: the environments used for testing the approach are visually clean -- in particular: the state information is sufficient to fully render / reconstruct the scene. This is likely not true for more realistic scenarios (eg think about autonomous driving where the commonly used state representations are certainly not sufficient to render all details in the environment -- the whole point is to reduce the amount of information in the policy's input). I wonder whether the proposed method would struggle with such environments since it constrains the "latent" variable of the prediction model to be equal to the pre-defined state while training it to reconstruct the full scene. If there is lots of detail in the scene that is not covered by the information in the state the model might struggle to reconstruct the scene properly.
- **requires action trajectories in the image data domain**: at least the version of the model that was experimentally validated requires access to action trajectories in the image domain (for the action reconstruction loss). Such action annotations might be hard to obtain in the real world -- baselines like CycleGAN do not require these since they learn the mapping purely from state and image data.

## Questions
- does the proposed approach need access to a differentiable behavior policy? I would think that just action samples would be enough and it would only need a differentiable policy on states (which is anyways available) --> however, the formulation in the paragraph before eq6 talks about a "differentiable behavior policy" so it would be good if the authors could clarify whether it is indeed needed
- how stable is the training of the approach? (Cycle)GAN approaches can be notoriously hard to train (see Appendix Sec D5) -- the proposed method also uses a GAN to minimize divergence from the prior. Some discussion on stability of this training or even a quantitative analysis of performance over a range of hyperparameters could help to show that this method might be easier to train than the GAN-based alternatives?
- when training the dynamics model online the simulator needs to be reset to the predicted state of the model --> how can value ranges be handled? ie what to do if the model predicts an invalid output state?
- what are the "numerical instabilities in Mujoco" mentioned in Section 4? maybe add a little more explanation?

## Suggestions to improve the paper
Addressing the weaknesses I listed above can help to improve the paper. In particular I would suggest to:
- clearly describe the baselines used
- show *why* the baseline is failing to make clear that this is not an issue of tuning
- include a GAN baseline that operates on short trajectory snippets instead of single states to see whether it is truly an issue of cross-modality or whether merely including dynamics information can help the simpler GAN baselines
- test on environments with (non-static) visual components that are not captured in the state information (eg add moving visual distractors to the Mujoco scenes which are not part of the state)
- ablate the action reconstruction component of the mapping loss to show that the method can work without the need for action annotations in the image domain
- tone down the contribution claims that talk about "transfer to real-world images" since the tested scenarios are far from real-world images

Some further, optional improvements:
- add a baseline that shows RL trained on images from scratch --> this can show how much performance we gain / loose by doing the domain adaptation vs training from scratch (even if we loose some performance it is okay since training from scratch might not be feasible in the real world)
- it seems that some of the differences to prior work require a better understanding of the technical details of the proposed method (particularly paragraph 4), it might be worth considering to move the related work section after approach before experiments
- as mentioned in my summary, it is a quite classic approach eg in robotics to separately train a perception module that maps images to states and then use it to work in the real world with policies that operate on state input (eg motion planners etc), ie perform cross-modal domain adaptation. However, they assume access to a paired dataset of [image, state] tuples, which is potentially hard to obtain. I think the related work section could benefit from adding a discussion about this.
- this sentence is not clear: "We first formulate the generation process of the real world and simulation" -- does this talk about how observations are actually generated in the real world or about the generation process of the model used in this paper? maybe reformulate for better clarity?
- Section 3.3 is a bit confusing, it only later became clear to me that training the separate dynamics model is optional, this could be emphasized a bit more.
- the dynamics mismatch experiment should be trained to full convergence (which seems to be ~10k steps for hopper), now it is only trained for 5k steps which equals only half-converged performance so it is unclear whether the mismatched runs will actually reach full performance

## Overall Recommendation
The proposed method is interesting and the paper certainly has merit. My main concern is that it is currently hard to judge the thoroughness of the experimental evaluation since it remains unclear how the baseline is implemented and why it fails. Therefore I cannot recommend acceptance at this point. If the authors can more convincingly show why prior work fails in the cross-modal case and how their method fixes that I am willing to increase my score.

---

> ### Author Response · Authors · 2020-11-16
> **Response to reviewer 2**
>
> We thank reviewer 2 for the helpful feedback and constructive comments.  Please find the response to your questions:
>
>
> Q1: Does the proposed approach need access to a differentiable behavior policy?
>
> We do not require the policy trained in the target domain (i.e. images) of any forms. We have updated the statement regarding this assumption.
>
> Q2: How stable is the training of the approach?
>
> Our approach is generally stable using different seeds. We will update the plotted figures of the training curve as soon as it is available. We think the training difficulty of CycleGAN mainly lies in the state-to-image part. The image-to-state GAN is relatively easier to train, as shown by our method and naïve GAN baselines.  We would like to investigate its sensitivity to hyperparameters in future experiments.
>
> Q3: How can value ranges be handled during the online training of the dynamics model?
>
> We adopt normalization to input and output in the training of the dynamics model to improve stability and avoid illegal outputs to our best. Since our method only needs 1-step prediction, the prediction error is limited, which is different from the compounding error problem in MBRL. If the illegal output still exists, we clip it to the valid range.
>
> Q4: What are the "numerical instabilities in Mujoco" mentioned in Section 4?
>
> The instability denotes the phenomena that MuJoCo engine outputs different $s\prime$ given the exact same (s, a) as input due to some of its inner inaccessible (random) states that cannot be reset by python interfaces. We have added a footnote to clarify it in the paper. Some further discussion can be found here [1].  In the two environments mentioned in the paper, the difference caused by this stochasticity exceeds a threshold so we do not train a DM.
>
> [1] https://github.com/deepmind/dm_control/issues/64
>
>
> Responses to weakness and suggestions:
>
> Q5: Is the generation process of real world and simulation modeling or ground-truth?
>
> It is how we model the generation process in the paper.
>
> Q6: More description of baseline GAN method and more experiments.
>
> We have updated the description of the GAN baselines in the experiment setting section. We also briefly describe the failed CycleGAN.  We also provide a quantitative error of mapped states of CODAS and GAN in Sec. D.3. The difference in error can partly explain the performance gap. We have also added a real training example in Sec. E to show that pure distribution matching using GAN can fail without extra constraints. We plan to run GANs based method on 4-consecutive images/states and will update the result as soon as it's available.
>
> Q7: More experiments on noisy environments.
>
> We are currently running experiments by adding some irrelevant objects in images and will update the result as soon as it is available. In fact, the chessboard pattern in the rendered images of MuJoCo is already a noise source to the mapping function and our method can tackle it well.
>
> Q8: What's the actual performance of the policy trained on images?
>
> We have updated the performance of the policy trained on states in Sec. D.1 of the new version. We are currently running experiments using image space by substituting MlpPolicy with CNNPolicy. We will update the result as soon as it is available.
>
>
> Q9: Requirements of action trajectories in the target domain.
> Currently, we view it as a limitation of our method. Wo are running ablation experiments to show how this may affect the performance. We would like to investigate how to relax such a requirement in the future work.
>
>
> Q10: Full-length results of dynamics mismatch.
>
> We are currently rerunning experiments and will update the result as soon as it is available.
>
> Q11: Overclaim of "real images".
>
> We have updated the description of all "real-world images" to "target domain images". We understand there is still a large gap before applying this method to actual sim2real applications. However, we believe this cross-modal setting is valuable in real-world applications of RL and our work provides an optimistic result to this setting.
>
> Q12: Reorganization of the article.
>
> Thank you for your suggestions on the organization of the article  (including more related works, placing related works after the method). We would polish our paper for the best readability.

---

> > ### Comment · AnonReviewer2 · 2020-11-20
> > **Doubt about Baselines**
> >
> > Thanks for your detailed answers!
> >
> > With the newly added descriptions of the baselines I am not convinced that the comparisons are fully fair. For a simple GAN-based domain mapping it is known that the kind of incorrect mappings as displayed in Fig.12 commonly appear, and certainly lead to worse results -- this is one reason that usually some form of cycle consistency is used to learn the mapping. The authors claim in the appendix that they were not able to train a CycleGAN mapping for the inter-domain mapping, but it still remains unclear *why* this did not work. Therefore it is also unclear whether this is an issue of tuning or whether the inter-domain mapping task indeed introduces more principled problems for CycleGAN methods.
> >
> > The proposed method further assumes access to actions in the image domain (which the baselines currently don't make use of). Baseline approaches could leverage these actions to further avoid the issues shown in Fig.12 by training an inverse model on the mapped representations and ensuring that its outputs match the true actions.
> >
> > Finally, the evaluation of baselines with dynamics information, ie with multiple stacked frames, is not yet provided.
> >
> > Therefore, I am not convinced that the comparisons in the experimental section are strong enough to prove the merit of the proposed approach.
> >
> >
> > **One more question about noisy environments**: does the state in the checkerboard environments include the position of the agent? If that was the case there would be a deterministic mapping between the state and the checkerboard pattern and it would therefore not serve as "a visual distractor not described by the state".

---

> > > ### Author Response · Authors · 2020-11-20
> > > **Response to doubts about baselines**
> > >
> > > Thanks for your additional comments and glad to know that most other concerns have been addressed by our previous update.
> > >
> > > The example in appendix E is a toy example to prove the existence of `'realistic but incorrect mapping'. As we mentioned in the intro and related work section, in same modality problems, extra constraints are relatively easy to design, but things are not trivial in cross-modal problems. So it serves as an intuitive example.
> > >
> > > Regarding your concerns with the  baselines,
> > >
> > > (1) we are running GAN with additional action information, i.e. GAN + action loss; Actually, the result of w/o RNN in the ablation study part can be viewed as an extension of this baseline (with extra recon loss + VAE encoder), so the result proves the effectiveness of CODAS (sequential structure + dm model). CODAS  does require action information and we would explore how to remove such a requirement in future works.
> > >
> > > (2) we are running GAN with 4-timestep tuples as inputs
> > >
> > > (3) we are trying to modify the images to add some static/moving objects irrelevant to the states. However, we think that the chessboard pattern is indeed a noise, in that,
> > >
> > >   (a) the state does not contain the absolute position of the agent so there is NOT a deterministic mapping
> > >
> > >   (b) we try to train a PPO from the images directly but the policy performs poorly in most environments, as the new results given in appendix D.2.
> > >
> > > (4) the reason why CycleGAN fails is that the training of state-to-image GAN is not stable (the loss fluctuates), while in CODAS, this part is done by a supervised loss. Another reason is that we have to move the id loss in the original CycleGAN because the inputs to two generators are no longer the same. The third is that some structures specially designed to stabilize GAN training cannot be easily applied here.  We have updated the analysis in the latest version.
> > >
> > > Thank you for your feedback.

---

### Official Review · AnonReviewer3 · 2020-10-27
**Interesting problem formulation; more clarity needed**

**Rating:** 4
**Confidence:** 4

**Review:**

Summary:

The authors pose a problem of learning a mapping when when a low-dimensional state simulation is given along with target image tragectories. The goal of the paper is to learn a mapping from image to state such that at test time the agent can directly use this mapping with a trained policy from the simulator to perform in the target domain. The contribution of this paper lies in its problem formulation and an algorithm named Cross-mOdal Domain Adaptation with Sequential structure (CODAS).

Pros:

This paper proposes a novel and interesting problem for cross-modal transfer.

I appreciate the paper shows the results when the dynamics of the source is slightly different from the target dynamics.

Cons:

Are there any assumption about the target data? Does data collection policy in the target domain resemble the pretrained source policy in the simulator in some way?
It seems to me that without this assumption q_\phi can be different from the groundtruth posterior p(s_t|o_t).
I am not sure if this is related to this sentence, "The policy \pi in source domains included in the second term does introduce a new assumption that real-world data are collected by a known behavioral policy." Do we need the policy in the target domain? If so, please clarify where this is used and mention the assumption in the beggining.

The experimental section needs clarity in baseline details. The GAN baseline is not explained in the paper making it difficult to interpret the comparative performance. Potentially, there should be a sequential version of GAN that generates tupples/sequences rather than just a single image [1].

What is the reward ratio? Please elaborate.
Does the groundtruth state-based (source) policy achieve the reward ratio of 1?

The dynamic mismatch should be investigated more. How differ can the dynamics be before things break? This is very important to understand whether this formulation can **actually** be applied to the real world settings.
I am curious to see what happen the states contain irrelevant information or be represented in a different way?

In equation 7, where does log p(\hat{o}_t|o_t) come from? Why do we not use the first term from l_D in equation 5 here?

Conclusion:

This paper proposes an interesting problem of cross-modality domain adaptation. However, there are a few questions to be addressed above to substantiate the potential of the proposed low-cost sim2real.


References:

[1] Learning Plannable Representations with Causal InfoGAN (http://papers.nips.cc/paper/8090-learning-plannable-representations-with-causal-infogan)

---

> ### Author Response · Authors · 2020-11-16
> **Response to reviewer 3**
>
> We thank reviewer 3 for the helpful feedback. Please see the responses below:
>
> Q1: What's the assumption of real-world data and real-world policy?
>
> Our method would benefit from a similar performance of policy trained on states and images, that is, the policy used for collecting data should have a similar cumulative reward as the pre-trained policy on simulated states. The claim that $p(s)$ and $q(s|o)$ matches holds under such an assumption. Similar assumptions have been introduced in previous works implicitly or explicitly as we state in Sec. 3.1.
> Our method does not require the policy trained in the target domain (i.e. images) in any form. We have updated the description for better understanding.
>
> Q2: What's $p(\hat{o_t}|o_t)$ in Eq. 7?
>
> $\hat{o_t}$ is the reconstructed observation from the mapped states $\hat{s}$. We use such a notation as we mentioned in the paper "For brevity, we will use $\hat{s}$, $\hat{o}$ to denote $s$, $o$ outputted by networks and omit the networks themselves." in Sec. 3.2.  Its concrete form is the same as the first term in Eq. 4.
>
> Q3: What is the reward ratio? Does the ground truth state-based (source) policy achieve the reward ratio of 1?
> Yes, it is the ratio of cumulative returns between the adapted policy and the original policy trained on states. The ground truth state-based policy achieves a reward ratio of 1. We have added a paragraph in Sec. 4 in the new version to clarify this metric.
>
> Q4: More results on dynamics mismatch.
>
> We have updated some new results on dynamics mismatched scenarios to investigate how robust our method is. We would like to point out, however, this experiment is designed to show that our method can still work under a mild difference, and it is not our focus. Domain adaptation under mismatched dynamics is another independent area. We think, combined with recent advances in this area, our method may perform better in sim2real applications.
>
> Q5: Performance of CODAS when states contain irrelevant information or are represented in a different way.
>
> We are currently running additional experiments by adding some irrelevant noises (e.g. randomly moving objects) in the rendered images and will update the result as soon as it is available. We also want to point out that the chessboard pattern in the rendered images is already a noise source to the mapping function and experiment results show that our method can tackle it well.

---

> > ### Comment · AnonReviewer3 · 2020-11-23
> > **Thanks for your response; a few more questions**
> >
> > Thank you for your effort in all the detailed responses -- clarifying my of my questions!
> >
> > Per baselines, I read your responses to Reviewer 1 and 2 as well. I still have a concern regarding more comparable baselines to help establish the results. Are the target data collected from expert policies, a mixture of experts, or neither? I would like to understand the quality of the offline dataset. Recently there are several SOTA methods on offline datasets that work better than BC method [1]. This will help us to understand if the simulators are really needed at all or there are other ways to extract a good policy directly from the offline data.
> > Another comparable baseline is to  do BC or GAIL with $\pi(a | q_\phi(o))$ -- the architecture that freezes the pre-trained state policy and only train the $q_\phi$ network from image to state.
> >
> > Per your response to Q5, I am also curious to see what happens if we apply a a transformation to the simulation states, e.g., one can represent the inverted pendulum by its world position coordinate, its joint angle to body, or its relative position coordinate. I would be excited to see if the proposed algorithm can work equally well regardless of this.
> >
> > Per your response to Q1, related to the point above, it seems that more assumptions are needed than just the reward matching, e.g., when there are many isomorphic solutions to $q_\phi$ maintaining the sequential structure might not provide the correct mapping.
> >
> > [1] Offline Reinforcement Learning: Tutorial, Review, and Perspectives on Open Problems https://arxiv.org/abs/2005.01643

---

> > > ### Author Response · Authors · 2020-11-23
> > > **Comparison to recent work in offline RL and other baselines**
> > >
> > > 1.Baselines
> > >
> > > We are running more experiments to prove the effectiveness of CODAS with baselines including (as our response to reviewer 2):
> > >
> > > a. we are running GAN with additional action information, i.e. GAN + action loss; Actually, the result of w/o RNN in the ablation study part can be viewed as an extension of this baseline (with extra recon loss + VAE encoder), so the result proves the effectiveness of CODAS (sequential structure + dm model). CODAS does require action information and we would explore how to remove such a requirement in future works.
> > >
> > > b. we are running GAN with 4-timestep tuples as inputs
> > >
> > > c. we are trying to modify the images to add some static/moving objects irrelevant to the states. However, we think that the chessboard pattern is indeed a noise, in that,
> > >
> > > 2.Differences to offline RL:
> > >
> > > Currently, the target domain dataset in collected by a (sub)-optimal policy (~0.9x return) because it's more controllable (policies of low return are much diverse and may violate our assumption).  We would explore how to relax such a constraint in future works. However, we want to point out the differences to offline RL:
> > >
> > > a. CODAS doesn't need per-step reward function in the target domain. Considering it's hard to design and calculate reward function in image domain, it can help reduce workload.
> > >
> > > b. To our best knowledge, few works in visual-input offline can work better than BC and its variants (BCQ, BEAR). See results on visual-input DM control suite in [1] (or put it another way, BC is a reasonable benchmark for offline learning). Some SOTA method like MOPO[2] and MOREL[3] require a model learning, and can hardly be applied in visual domain.
> > >
> > > c. The goal of our method is to use as few data as possible in the target domain with the help of a simulator. Though CODAS is currently trained using 600 trajs, we believe this can be improved in the future work, while Offline RL generally requires far more data.
> > >
> > > 3.Different state representation and isomorphic mapping:
> > >
> > > We think a different state representation would not affect the learning much because the difficulty of learning a mapping function is the same to the NN. The only influence may be the learning of DM. However, if the state is well designed to capture the dynamics info, this would not affect much neither.
> > >
> > > We add policy loss to solve the case as you mentioned. If the learned state is wrong but lead to same action, then it's not a problem. We agree that there may be some rare cases that are not distinguishable even with the sequential structure, however such a problem is much rarer. As our experiments shows, CODAS do perform better than vanilla GAN in a practical cross-modal domain adaptation problem.
> > >
> > > Thank you for your feedback.
> > >
> > > [1] RL Unplugged: Benchmarks for Offline Reinforcement Learning. https://arxiv.org/pdf/2006.13888.pdf
> > >
> > > [2] MOPO: Model-based offline policy optimization https://arxiv.org/abs/2005.13239
> > >
> > > [3] MOReL : Model-Based Offline Reinforcement Learning https://arxiv.org/abs/2005.05951

---

### Official Review · AnonReviewer1 · 2020-10-28
**New topic in RL, but experimental results are missing**

**Rating:** 5
**Confidence:** 3

**Review:**

## Cross-modal Domain Adaptation for Reinforcement Learning
### Summary
The authors propose CODAS, a domain adaptation algorithm for transferring policies learned in state-space to the image domain. They learn a variational RNN to map image inputs to state space with an adversarial loss to ensure that the distributions of the mapped states and the original states are matched. Their experiment results show improvement over one behavior cloning baseline and one GAN domain adaptation baseline.

Overall, the paper is clearly written, and the topic is new and relevant to the field.

### Strength
- The first cross-modal adaptation for the RL agent to the best of my knowledge.
- No paired data is needed.
- Interesting DM model.
- Good improvement over the baselines considered in the paper.


### Weakness
- Insufficient experimental results. Specifically:
   1. What's the performance of the policies trained in the original state-space?
   2. What's the performance of the policies trained directly in the image space?
   Knowing these numbers gives us a better idea about how well the adaptation works.

- Unclear description of GAN baselines. Did I miss something? Can't find a description in the appendix as well.

- Need human expert policies to collect image input. It is unclear how this component may affect the final performance of CODAS.

- "In this paper, we propose Cross-mOdal Domain Adaptation with Sequential structure (CODAS) that learns a mapping function from images in the target domain (real world)" (page 2): this is misleading. The target images are not really from the real-world. The images are rendered in simulation, with no realistic perturbations.

### Minor Comments
- Notation: why refer to Eq in the appendix instead of the main paper?

---

> ### Author Response · Authors · 2020-11-16
> **Response to reviewer 1**
>
> We thank reviewer 1 for the helpful feedback. Please see the responses below:
>
> Q1: What's the actual performance of the policy trained on states and images?
>
> We have updated the performance of the policy trained on states in Sec. D.1  in the new version. We train the policy using PPO from stable-baselines and the results correspond to previous reports. We are currently running experiments using image space by substituting MlpPolicy with CNNPolicy. We will update the result as soon as it is available.
>
> Q2: Unclear description of baseline GAN methods.
>
> We have updated the description of the GAN baselines in the experiment setting section. We also briefly describe the failed CycleGAN. Basically, it follows a conventional GAN training procedure in the domain adaptation area by using two unpaired datasets and the same network structure (if possible) as our method.
>
> Q3: What's the assumption of real-world data and real-world policy?
>
> Our method would benefit from a similar performance of policy trained on states and images, that is, the policy used for collecting data should have a similar cumulative reward as the pre-trained policy on simulated states. Similar assumptions are introduced in previous works implicitly or explicitly as we state in Sec. 3.1. We use expert policy in the experiments because it's more convenient and more standard for comparison. We would like to investigate in future works if the policy for data collecting is much different from the policy we want to adapt to.
>
> Q4: Overclaim of "real images".
>
> We have updated the description of all "real-world images" to "target domain images". We understand there is still a large gap before applying this method to actual sim2real applications. However, we believe this cross-modal setting is valuable to real-world applications of RL and our work provides an optimistic result to this setting.
>
> Q5: Wrong reference of Equation.
>
> We have updated the reference to refer to the equation in the main paper.

---

> > ### Comment · AnonReviewer1 · 2020-11-20
> > **Thanks for addressing my questions.**
> >
> > The newly added blue texts and tables in the appendix do help clarify my previous problems.
> > Thanks for all the efforts.
> >
> > Here are a few more questions:
> > #### Assumption about real-world data and policy
> > I have read the response about this to both me and reviewer 3. In your response, you mentioned that:
> > 1. 'the data collecting policy should have similar performance to pre-trained policy on simulated states.'
> > 2. 'Our method does not require the policy trained in the target domain in any form.'
> > These two points look contradictory to me. How can you get a data collecting policy with performance similar to a state-space policy without training? Doesn't 1. mean that you still have to train an image-space policy (although this policy is not exactly used for training your CODAS)?
> >
> > Additionally, in your original manuscript, you mentioned that $\pi_\text{behavioral}$ is a human expert policy. Isn't this a harder constraint than 2.? I also notice that you revise the bit about human expert policy in the updated version. Does that mean you are no longer using this expert policy for your new results (although the outcomes all look identical to me)?

---

> > > ### Author Response · Authors · 2020-11-20
> > > **Clarification of Assumption about real-world data and policy**
> > >
> > > Glad to know that most of your questions are properly answered.
> > >
> > > 'the data collecting policy should have similar performance to pre-trained policy on simulated states' `does not require a policy trained on target domain. A performance metric to evaluate the collected dataset is enough, e.g. rewards or success rate of certain tasks. We can train another policy in the source domain until reaching a similar performance as the behavioral policy in the source domain. This is consistent with our response to reviewer 3 that  CODAS doesn't require a differentiable or non-differentiable policy in the target domain. To summarize, we only need data in the target domain but not policy.
> > >
> > > To address your additional concern, we do not use this policy from the very beginning. We merely update the description since the previous ones may cause some misunderstanding. All results are the same.
> > >
> > > Hope that this answer could address your concern.

---

### Official Review · AnonReviewer4 · 2020-11-02
**The paper explores cross modal adaptation to map system state (available in sim) to observations (available in real world) in RL setting**

**Rating:** 5
**Confidence:** 3

**Review:**

This paper explores learning the mapping from state to sensor observations as part of RL policy learning. This is a relevant problem to enable sim-to-real transfer.  The paper explores cost terms that have been used in GANs to align distributions between the state and observation. The paper shows experiments in simulation, where the observation is images of the robot in the environment. I have some concerns  which are listed below,

1. The paper only shows transfer from the robot's state to images of the robot in the environment. The state of a system might not sufficiently capture high fidelity information from observation. E.g., can this method reconstruct uneven terrain from only observing the robot's pose and other low dimensional representations in state? It's unclear to me how the method can transfer to work with observations on real systems.

2. All the chosen environments do not involve a second object to interact with. E.g., can the method work on the OpenAI cube rotation task where the state could contain the object pose and the observation might not. It's also shown by other researchers that walking doesn't require high fidelity observations~[1] such as vision so the effectiveness of the method to transfer policies trained on state to observations is unclear. Is the method overfitting to recorded trajectories? one good test would be to deploy the policy in a different environment(increased slope of the floor?) that wasn't part of the recorded trajectories but was part of the full state policy.

3. It would be interesting to discuss how this work could leverage or compare against multi-modal representation learning in robotics~[2].


References:
[1] Lee J, Hwangbo J, Wellhausen L, Koltun V, Hutter M. Learning quadrupedal locomotion over challenging terrain. Science Robotics. 2020 Oct 21;5(47).
[2] Lee MA, Zhu Y, Srinivasan K, Shah P, Savarese S, Fei-Fei L, Garg A, Bohg J. Making sense of vision and touch: Self-supervised learning of multimodal representations for contact-rich tasks. In2019 International Conference on Robotics and Automation (ICRA) 2019 May 20 (pp. 8943-8950). IEEE.

---

> ### Author Response · Authors · 2020-11-16
> **Response to reviewer 4**
>
> We thank reviewer 4 for the helpful feedback. Please see the responses below:
>
> Q1: The state of a system might not sufficiently capture high fidelity information from observation.
>
> Our method focuses on using the mapped states, while the reconstruction term is an auxiliary loss that enhances the training process. As shown in the ablation studies in Sec. 4.2, the reconstruction term, GAN, and the specially designed network architecture all contribute to the overall performance. It's worth noting that we also allow $\hat{o_t}$ to be dependent on $o_{t-1}$ so that some irrelevant objects in the images do not affect the training much because they can be reconstructed in an autoregressive way. In fact, the chessboard pattern in the rendered images is already a noise source to the mapping function and experiment results show that our method can tackle it well. We are currently running additional experiments by adding some irrelevant noises (e.g. randomly moving objects) in the rendered images and will update the result as soon as it is available.
>
>
> Q2: Why use MuJoCo as the experiment environment?
>
> In the RL community, MuJoCo is one of the most widely adopted benchmark environments for robotics control, e.g. in [1-4]. We are currently planning to extend our method to work in more complex environments, e.g. OpenAI Cube Rotation task and real-world robot grasping task in future works. We agree that these complex tasks, to some extent, require more engineering work. For example, in the OpenAI Cube task, the agent needs an oracle policy to solve the task. We understand there is still a large gap before applying this method to actual sim2real applications and there indeed exists cases where such a method is not applicable. However, as we stated in the paper, we believe this cross-modal setting is valuable to real-world applications of RL, and our work provides an optimistic result to this setting.
>
>
> Q3: Does our method overfit to the recorded trajectories?
>
> We use a stochastic policy to collect the datasets and do performance evaluations. The adapted policy cannot perform well in such a case if the mapping function is overfitted. The extended experiment results on environments with dynamics mismatch in Sec. D.5 further prove that there is unlikely an overfitting in our method.
>
> Q4: What's the connection between this method and some multi-modal works?
>
> We believe multi-modality representation learning is a different field from what our work discusses. In the multi-modality setting, the input comes from different modality simultaneously, while our method focuses on mapping the input from one modality to the other, and the policy runs on only one modality. However, we think there may be some common techniques shared between these two areas and we would like to investigate it in future works.
>
> [1] Kuno Kim, Yihong Gu, Jiaming Song, Shengjia Zhao, Stefano Ermon. Domain adaptive imitation learning. In ICML, 2020.
> [2] Bradly C. Stadie, Pieter Abbeel, Ilya Sutskever. Third Person Imitation Learning. In ICLR, 2017.
> [3] Irina Higgins, Arka Pal, Andrei A. Rusu, Loic Matthey, Christopher P Burgess, Alexander Pritzel, Matthew Botvinick, Charles Blundell, Alexander Lerchner. DARLA: Improving Zero-Shot Transfer in Reinforcement Learning. In ICML, 2017.
> [4] Brahma S. Pavse, Faraz Torabi, Josiah Hanna, Garrett Warnell, Peter Stone. RIDM: Reinforced Inverse Dynamics Modeling for Learning from a Single Observed Demonstration. IEEE Robotics Autom. Lett. 5(4): 6262-6269 (2020).

---

> > ### Comment · AnonReviewer4 · 2020-11-20
> > **Observations in realistic environments unclear**
> >
> > Thanks for your clarification. Some points:
> >
> > Q2: There are some dexterous tasks in Mujoco as well, as shown here: https://sites.google.com/view/deeprl-dexterous-manipulation
> >
> > From the current set of experiments, it's unclear how the approach will work on more realistic settings, especially when the environment has more varied dynamic behavior:
> >
> > 1. One example I can think of is the "In-hand manipulation" task from the above url. If the state is defined as the object pose and the hand state (joint angles, velocity), is it possible to reconstruct images (assuming image is the observation space)?
> >
> > 2. Another example is walking in uneven terrain or stairs, where local terrain is part of the state (might be unavailable in existing Mujoco environments).

---

> > > ### Author Response · Authors · 2020-11-23
> > > **Reconstruction of high fidelity images**
> > >
> > > Thank your for your additional comments.
> > >
> > > We want to first rephrase your questions (please check if we understand it correctly) and then answer it.
> > >
> > > Q: Does CODAS need a high-fidelity reconstructed image?
> > >
> > > No. The reconstruction loss in the objective function is used to help the learning of states. All we care about is the quality of the learned state because they are fed into the downstream policy. Even if the reconstruction wasn't 100% accurate. it's still possible to learn a good state. See illustration in Fig.4, where the reconstructed images are somewhat inaccurate but the re-rendered images are accurate.
> > >
> > > Q: Can CODAS work in more complex environments?
> > >
> > > We plan to to so in future works. As we stated in previous responses, there may be some engineering concern in such environments. However, we think CODAS itself is a self-contained paper that proposes a new paradigm of low-cost sim2real training and would help some real-world application of RL.

---

### Author Response · Authors · 2020-11-24
**Summary of revisions and final remarks**

Thank all reviewers for your constructive feedback and suggestions. We have updated the final revision accordingly. The changes include

(1) we have rephrased the contribution to avoid using 'real world' in the paper;

(2) we have added training curves of policy trained on states and images;

(3) we have fixed some statements for better readability and understanding;

(4) we have added a toy example to demonstrate the existence of 'realistic but incorrect' mapping.

(5) we have added experiments on GAN with policy loss as new baselines;

(6) we have added multiple seeds for CODAS to demonstrate its stability;

We understand there is still a large gap before applying this method to actual sim2real applications. However, as we stated in the paper, we believe this cross-modal setting is valuable to real-world applications of RL, and our work provides an optimistic result to this setting.  We think that this paper is self-contained and hope that it can inspire some new researches in this setting. We will extend this idea to more complex applications in future works.

---

### Decision · Program_Chairs · 2021-01-07
**Final Decision**

**Decision:**

Reject

**Comment:**

The reviewers generally appreciated the problem statement and topic of the paper, but raised concerns across the board about the empirical evaluation. Since the paper is largely experimental in nature, a compelling experimental evaluation is important. Reviewer concerns generally centered around: (1) the relative simplicity of the evaluated tasks; (2) somewhat questionable baselines. While the authors provided responses to some of these concerns, after discussion the reviewers generally still felt that these issues were rather severe, and preclude publication at this time. I would encourage the authors to take this feedback into account in improving the empirical evaluation in the future.